# Caregivers’ Experiences with School–Work Transitions for Their Children with Disorders of Intellectual Development

**DOI:** 10.3390/ijerph20031892

**Published:** 2023-01-19

**Authors:** Veerle Garrels, Hanne Marie Høybråten Sigstad

**Affiliations:** 1Department of Vocational Teacher Education, Oslo Metropolitan University, 0130 Oslo, Norway; 2Department of Special Needs Education, University of Oslo, 0318 Oslo, Norway

**Keywords:** disorders of intellectual development, intellectual disability, employment, school–work transition, parental stress, support needs, labour market participation

## Abstract

During the period of school–work transition, caregivers of young adults with disorders of intellectual development (ID) often play an extended and leading role in supporting their children. This article explores caregivers’ overall experiences with their children’s school–work transition. Ten qualitative in-depth interviews were carried out with eleven parents/guardians of ten young adults with disorders of ID. Through reflexive thematic analysis, the following themes emerged: (i) varying degrees of preparation for employment during school years; (ii) the experience of transition collapse; (iii) struggling to navigate the system; (iv) caregivers’ ambitions and high expectations; and (v) positive meetings with professionals. All caregivers in our study had clear ambitions about employment for their children, and they supported them by advocating for their rights and by collaborating as best as possible with the support system. However, their experiences bring to light how the transition process often appears random and without an overarching implementation strategy. The overall picture of the transition process is a time of concern and stress for caregivers, with room for improvement in most areas.

## 1. Introduction

Disorders of intellectual development are a group of conditions characterized by impairment in intellectual and adaptive functioning [1]. The World Health Organisation’s 11th revision of the International Classification of Diseases and Related Health Problems (ICD-11) has suggested the term “disorders of intellectual development” (in this article abbreviated to “disorders of ID”) to replace the diagnostic terms “intellectual disability” and “mental retardation”. People with disorders of ID often require individualised supports throughout their lifespan, and especially during times of transition, they may need more intensive support [1]. According to the ICD-11, young adults with disorders of ID may experience challenges related to assuming adult roles, such as employment. However, ICD-11 also states that people with mild disorder of ID, i.e., with intellectual and adaptive functioning that is approximately two to three standard deviations below the mean, can generally achieve employment if appropriate support is provided [1]. Yet, research consistently finds that people with disorders of ID are only marginally represented in the labour market, and internationally, their participation in employment remains alarmingly low [2]. When people with disorders of ID are excluded from employment, this may result in several negative consequences for them, such as poorer mental and physical health [3,4], a lack of structure in everyday life [5], fewer social relationships [6], and poorer overall quality of life [7].

For young people in general, emerging adulthood is typically a period of possibilities, characterised by a move towards independence and self-sufficiency [8]. It is a time in which aspirations for long-term romantic relationships, career development, and moving into a place of one’s own tend to materialise. Consequently, the transition into adulthood usually implies a change in the relationship between parent and child, and parents may shift roles from a hands-on caregiving role to a more supportive, guiding role [9]. Parents may experience this transition as worrisome, and they may need time to adjust to the newly gained adult status of their children [10]. Yet, children’s transition into adulthood may also lead to enhanced wellbeing and improved mental health for parents [11]. However, for parents of children with disorders of ID, this period of emerging adulthood is more often associated with increased levels of stress and uncertainty [12].

Statistically, parents of children with disorders of ID are at a higher risk of depression and anxiety compared to parents of typically developing children, due to various stressors, such as increased caregiver demands and financial strain [13]. When children with disorders of ID grow up, parents’ concerns may take on new dimensions, and the psychological impact on parents may even be intensified [14]. During the transition period from school to working life, parental stress may be aggravated because of the numerous barriers that people with disorders of ID experience when trying to gain access to the labour market. For parents, it may be upsetting to see their child struggle with the transition to employment and adulthood, and parental wellbeing may be especially impacted if support systems are difficult to access [15,16]. Still, according to some studies, e.g., [17,18], parents of emerging adults with disorders of ID may also experience relatively high levels of life satisfaction and family quality of life and feelings of fulfilment, growth and gain during this transition period. These positive sentiments seem to be more present when the child’s support needs are lower [17].

While parents of typically developing children may shift roles from caregivers to supporters who stand on the side-line, parents of emerging adults with disorders of ID often play an extended, leading role in the school–work transition of their children [19]. For people with disorders of ID, transitioning from adolescence into adulthood implies transitioning from childhood and adolescent services into adult services, and parents may often take on a coordinating role in this process [20]. Furthermore, their extended role may include helping the young adult with job searches and preparing for job interviews [16], being advocates for their adult children [12], and involvement in goal setting for post-school life [21]. Thus, the transition from school to employment for young people with disorders of ID is usually not a solo process, but instead, it is a joint project in which parents, school, and other agencies also engage [19]. For parents, it may seem that it is not the extended role and collaboration with multiple instances in themselves that contribute to enhanced feelings of stress during transition. However, insufficiently supportive adult services and what parents describe as “battling against services, lacking trust in services and incompetent professionals” seem to be a cause of negative transition experiences [12] (p. 45). Moreover, since labour market participation amongst people with disorders of ID is persistently low, parents may also experience stress and anxiety related to the limited employment opportunities that are available to people with disorders of ID [22]. In this context, it is important to expand the research agenda in order to gain a greater understanding of the social–ecological constraints that parents of children with disorders of ID face and to identify the resources that they need to meet the everyday challenges that they experience [23].

Thus, previous research informs us that caregivers may experience the transition from school to employment for their adult children with disorders of ID as stressful, but the transition may also be associated with more positive feelings. However, few studies have explored in-depth which factors seem to influence the transition experience and how parents of young people with disorders of ID support their offspring during the transition process.

This article explores how caregivers of young adults with disorders of ID experience the transition from school to work for their children. Through in-depth qualitative interviews, the study aims to investigate the following research question:


*What characterises caregivers’ overall experience of their children’s school–work transition?*


The following sub-questions were defined to shed light on the study’s main research question:(i)Which challenges do parents encounter during this transition phase?(ii)Which factors seem to contribute to a positive transition experience for both the caregivers and the young adult with disorders of ID?

Findings from this study may inform researchers, professionals and policy makers about challenges and barriers that caregivers experience during their children’s school–work transition. In addition, this article highlights factors that may contribute to a more positive transition experience for both caregivers and their children with disorders of ID. This information may inspire support systems, such as schools, supported employment companies, etc., about how to collaborate more effectively with caregivers towards the common goal of employment for all.

## 2. Materials and Methods

### 2.1. Study Design

This study is part of the project “Effective school–work transitions for students with mild intellectual disability”, funded by the Norwegian Research Council (project number 301510). In the qualitative part of this project, we used a case study approach with in-depth interviews to investigate the characteristics of a successful school–work transition for young adults with disorders of ID. The case study consisted of ten cases, where each case was centred around a young employee with a disorder of ID who had successfully managed the transition from school to competitive and integrated employment with support. Furthermore, each case also included the parent(s)/guardian, teacher, employer, job coach, and sometimes also social worker and pedagogical–psychological counsellor of the young person around whom each case was built up.

The present article summarizes the part of the qualitative case study that relates to the school–work transition from a caregiver perspective. Analysis of in-depth interviews with caregivers of young people with disorders of ID who had been successful in entering the labour market after finishing upper secondary school forms the basis for this article.

### 2.2. Ethical Considerations

The Norwegian Centre for Research Data (NSD) approved this study (approval number 380880). Initially, informed consent was obtained from the young adults with disorders of ID, before we contacted their parents/guardians for an interview. Prior to the interview, participants received an information sheet about the study and a consent form. The information sheet emphasised that participation was anonymous and voluntary and that research participants could withdraw from the study whenever they wished to do so without stating a cause.

### 2.3. Recruitment and Sample Description

For this part of the study, we first contacted young people (aged 20–25) with disorders of ID who had managed a successful school–work transition. In order to get in touch with them, we contacted different stakeholders in south-eastern Norway, such as supported employment agencies and schoolteachers. After obtaining informed consent from ten individuals with disorders of ID, we contacted their parents or legal guardian and invited them for an interview. In seven cases, one of the parents took part, and in one case, both parents participated. In two cases, a guardian took part. In total, this means that our data material represents eleven parents/guardians (six female, five male) of ten young adults with disorders of ID. All participants were recruited in south-eastern Norway.

### 2.4. Data Collection

The interviews were either performed in the caregiver’s home or alternatively at a public place such as an office or library. The interviews were mostly carried out with two researchers present. We used a semi-structured interview guide with predefined ideas, and more detailed follow-up questions were continuously developed, based on participants’ responses. The interviews were centred around aspects that participants identified as important for their experience of their children’s school–work transition. Examples of the predefined questions that we asked were: *Can you tell us about your child’s school career; How was career planning incorporated into his/her individualised education plan*; and *What would you identify as pivotal for your child’s successful school–work transition?*

During the interviews, it soon became clear that for many participants, recounting their experiences regarding the transition process for their children with disorders of ID was emotionally taxing. When thinking back to what they had been through, some of the parents became emotional during the interviews. As researchers, we tried to acknowledge the participants’ feelings and to normalise their reactions of upset after the struggles that they had encountered. When participants showed signs of distress, we were careful not to hurry on with the interview, but to allow time for them to rebalance themselves. Furthermore, we were mindful of letting the participants steer the interview forward, and we checked whether participants were still comfortable with sharing their stories. All the participants confirmed that they wanted to share their experiences so that they could make an active contribution to improve the school–work transition for others in the same situation.

The average duration of the interviews was 58 min (range 38–74 min). Unfortunately, one of the ten recordings failed, and therefore, the interview was summarised in written text soon after completion. The other nine interviews were successfully audiotaped and transcribed verbatim. With ten interviews, data saturation was achieved.

### 2.5. Data Analysis

Interview data were analysed using Braun et al.’s [24] six-phase approach to reflexive thematic analysis. During the first phase, both authors read the transcribed interviews individually to become familiarised with the data. At this point, we used an inductive data-driven process to identify meaningful quotations in the interviews, and these were manually highlighted. During the second phase, the highlighted text fragments received a code, i.e., a key word or phrase, that captured their content. Then, during the third phase, we compared our individual coding and constructed subthemes based on the coded interview fragments. During the fourth phase, subthemes were further collated and pruned and then grouped into five main themes that could illustrate the research questions of this study (see Table 1 for an example of the data analysis procedure).

During the fifth phase, the names of the themes were decided upon, so that the essence of each theme was adequately conveyed by its name. During the sixth and final phase, the results of the data analysis were written down in this article and discussed in light of the existing knowledge base. At this point, the selected interview fragments were translated from Norwegian into English by the first author.

## 3. Results

The following themes were identified in our data set: (i) *varying degrees of preparation for employment during school years*; (ii) *the experience of transition collapse*; (iii) *struggling to navigate the system*; (iv) *caregivers’ ambitions and high expectations;* and (v) *positive meetings with professionals*. The first three themes relate to our question of which challenges that caregivers experience during the transition process, while the fourth and fifth theme point out success factors. Within each theme, we describe several subthemes (see Figure 1) and illustrate these with relevant interview fragments underneath.

### 3.1. Varying Degrees of Preparation for Employment during School Years

For most caregivers in our study, thinking back to their children’s years in upper secondary school made them relive a period characterised by frustration and even despair. Participants seemed generally malcontent with the school’s efforts to prepare children with disorders of ID for working life. Several of them witnessed that teachers were problem-oriented and that they had low expectations on behalf of the students:
*Richie’s school was not work-oriented at all. The meetings we had with them were not very nice, because I got so frustrated when they only saw limitations. They focused a lot on medication and problems. No, they were not at all proactive.*

Several participants also experienced that teachers were too laid back when it came to stimulating students’ cognitive development and learning. Some caregivers explained how school focused on teaching skills that their children already possessed, indicating an inability to provide students with individualised education. Furthermore, ambitions to work systematically to teach students important skills for working life often seemed to be lacking:
*My experience all along is that the school has really nice plans. But reality turns out to be totally different. They [teachers] try and do their best and follow up, but in the end, when they’ve been in school for five years, it feels like a waste of time. It’s not goal oriented. These kids need something that is more work oriented.*

Yet, even when schools made plans for preparing students for working life and securing a smooth school–work transition, intentions and everyday reality were not always aligned. Several participants had experienced that schools had the objective to provide an education that was goal oriented and focused on future employment, but in many cases, schools did not deliver on their promises. As Danny’s Mum explains:
*Danny’s IEP had some really nice goals, like “Trying out possibilities in different workplaces”. But in reality… Even if they mean well and try to follow up, there’s a lot of things going on. Teachers get sick, they move, they quit… And then nothing happens. The good intentions were there all along, but nothing concrete was coming out of it.*

In certain cases, caregivers also felt that they were handed over certain responsibilities that in principle were the school’s mandate. One father related how he took over the responsibility for teaching his son important skills and competencies, since school did not seem to deliver. Another parent witnessed how she was prompted by the school to find an opportunity for job training for her son:
*During the first year, school informed us that we had to find some sort of job training for Walter. That task was left to us parents. We got a very clear message from school, saying that it was very limited what kind of support that they could provide.*

However, a few of our participants also had very positive experiences with the school’s efforts to prepare their children for future employment. Particularly, caregivers appreciated it when schools could provide their child with job training in a company, which they saw as beneficial for their child’s development and later job opportunities:
*I would say that school has been the most decisive factor for her getting a job. They made it possible for Lisa to get job training in a childcare centre. They allowed her to be there first one day a week, and then two days. I think that helped her a lot to get a job there after school.*

Thus, job training while still at school was highlighted as very important, so that employers could get to know the person with a disorder of ID and see what he or she could contribute with as an employee. Yet, not many of our participants had this positive experience for their child.

Participants also valued it when teachers saw their children as individuals and when they managed to develop a study program that was adapted to the student’s strengths and needs:
*Eddie’s teacher was amazing. School came up with a very nice tailormade program, and they took into consideration his needs.*

Thus, in certain cases, caregivers experienced that schools and teachers played an important role in their children’s school–work transition and that teachers worked effectively to prepare their children for working life. However, the dominant impression of most participants was a school that was unable to work systematically and goal-oriented towards this transition. As such, caregivers experienced first and foremost that school had left their children with disorders of ID underprepared for future employment.

### 3.2. The Experience of Transition Collapse

Despite that all participants in our study had children who had managed the transition from school to working life relatively smoothly, several of them referred to it as a period of long and worrisome waiting before their children obtained supported employment. As one mother told us:
*We left him behind when we ourselves were going to work; he was just sitting at home, completely left to himself during June, July, August, and September. Four and a half months maybe.*

While short periods without any structured activity or daytime occupation between school and employment may be quite normal for young adults in general, parents of children with disorders of ID experienced feelings of stress and unease when there was no job waiting for their children after upper secondary school. Several participants expressed fear that their children would lose important skills during this time of inactivity, and they were also concerned about what the loss of a daily routine would mean:
*He didn’t have anything to go to. From having had a very structured day, which is really important, to have some place to go to, something to get up for in the morning, to have a routine… You know, the kind of structure that you spend a lot of time building up. And then that is all gone. Summer holidays arrive, and his entire life becomes a summer holiday, and he gets enormously isolated. Because then there is only leisure time, and he’s sitting there watching YouTube videos on his mobile phone.**(Walter’s Mum)*
*I saw him sitting there. I remember that I was on my way to work, and he was sitting on the couch. And I thought “Oh God, this is not good…”.**(Danny’s Mum)*

For some participants, the transition was complicated by a lack of coordination. Typically, different instances were involved during the transition process, but these did not always seem to communicate and collaborate well together:
*It was really difficult to get everyone involved to agree on what kind of plan we were going to have for Simon. […] Different people need to work together. And even then, there is no guarantee for success, so you need to have some luck as well.*

One parent experienced the lack of coordination as an almost surrealistic display of bureaucracy, where it was often difficult to get in touch with those involved, and even more complicated to obtain different instances to speak and work together towards a common goal:
*I was waiting and waiting, but nobody contacted us. I called the social services, but they claimed that the supported employment agency was responsible for following up. So, I got in touch with the supported employment agency, and they said that they were waiting for the social services to give them the green-light. […] In the end I called the social services and said “Even if you guys think that he’s under supported employment, supported employment is sitting and waiting for thumbs-up from you. Could any of you talk together? Because I’m just telephoning back and forth. Could you talk together, please?”*

Moreover, as their children turned 18, caregivers needed to obtain a power of attorney to communicate with the support system, and several participants experienced this as an obstacle when trying to support their children during the transition. Fortunately, the experience of transition collapse was not shared by all our participants and a few related positive experiences where their children transitioned seamlessly from school to employment:
*We persuaded Simon not to leave school until he had a job or prospects of a job. So, we succeeded with that—and we were lucky, of course. And then the transition was actually quite uneventful. He quit school one day and started working the next day.*

Thus, for most participants in our study, the transition period between school and employment represented a time of uncertainty and anxiety, without any guarantees of when a job would become a reality for their children. This fact of not knowing what the future would bring clearly stressed them. In addition, they struggled with trying to coordinate the transition process, a task that was often complicated due to bureaucracy and a lack of collaboration between services. For those who managed a smooth transition, some luck seemed to have played a role.

### 3.3. Struggling to Navigate the System

Most of the participants in our study found it hard to find the information that they needed about the support systems and provisions that are available to people with disabilities and their families, and they highlighted this as an area for improvement:
*For parents, who are responsible for these children, it is very important that they get the right information. So, the whole information flow could be improved a lot. And the entire support system, especially the schools, they could do a lot better.*

Even parents who defined themselves as resourceful described challenges with finding their way in the support system, and this seemed to pose an extra burden on their shoulders. Their expectations towards themselves of being able to manage were not always met, and this could sometimes result in a low sense of self-efficacy:
*We are not living encyclopaediae, you know. Now you have to remember this, and now you have to do that. There’s an incredible number of things that you need to know when you have a child with a disability, but they don’t come with a user manual.*

Despite what seemed like a common experience of support systems working against them rather than with them, most participants found that it was easy for their children with disorders of ID to receive a disability pension. Some caregivers told us about an “express treatment” from the social services with a speedy eligibility assessment:
*Interviewer: How about the disability pension? Was that something that came easily?*
*Mother: Yes, as a matter of fact! I was flabbergasted. I had never thought of applying for financial benefits. But I was very happy that someone cared and told me that there was an arrangement that he could benefit from.*

Yet, this fast process of granting a disability pension was not always what caregivers preferred. Some were concerned that it would hamper their children’s possibilities to participate in the ordinary labour market, as it could signal that there were no expectations that someone with a disorder of ID could manage employment. As Nick’s guardian explained:
*It feels like being trampled on. He hasn’t even signed the documents for a disability pension yet, but it’s already been taken care of.*

From the interviews, it was clear that caregivers generally had a hard time navigating the support system. Their descriptions painted an image of support systems characterised by vast bureaucracy, which made it difficult to find out whom to contact and where to find information. Contrary to participants’ expectations, receiving a disability pension was a more straightforward process. While most caregivers were grateful for this economic safety net for their children, some were also worried that it would hinder their chances of entering the labour market.

### 3.4. Caregivers’ Ambitions and High Expectations

All caregivers in our study emphasised the importance of having high expectations and ambitions for their children with disorders of ID. Participants expressed a clear vision of the future that they envisaged for their children, and ordinary employment seemed a natural part of that picture: “Our goal has all the time been to get Simon into ordinary work. Not a special workplace where he would be stigmatised and where there is no regular work environment.” Thus, the caregivers in our study seemed to expect that their children could perform adequately in a regular workplace and that employment in such workplaces was within reach for them. Participants also recounted conversations that they had had with their children about these expectations for future employment, sometimes from an early age on:
*We’ve always pushed her. Already in primary school, Emma said “Mum, I don’t think that I can work in a shop”. So, I asked her why she wouldn’t be able to do that, and she said “Do you think that I should be handling money? That’s not going to end well!”. But then I said: “But you’ll learn all that, you know.”*

Thus, for the caregivers in our study, their children’s transition from school to working life was considered a logical step towards adulthood, and they seemed resolved that this was not negotiable:
*I will not have my kid sitting at home while she could be doing a job. I will not have her just sit at home and watch TV all day. I’ve been very determined about that. We’ve both been very determined about that, me and my husband.*

Caregivers in our study did not see the diagnosis of disorders of ID as an impediment to ordinary employment in a competitive workplace, and they were adamant about seeing ability before disability. Caregivers seemed to have a firm belief that their child could be an asset to society and the workforce:
*They have a diagnosis, but they have so many qualities too. Everyone can contribute. And that is really important when it comes to the current labour market situation, with the lack of manpower these days.*

Furthermore, some participants related how they had been met with low expectations and bleak predictions about their child’s future development and employment possibilities, and proving these voices wrong seemed like an additional incentive for aiming high:
*He needs to be able to take care of himself, and he has done so from day one. Easy as that. There were some people who didn’t believe that he would be able to do that. Proving the people from the health services wrong has been a big thing.*

Some of our informants also emphasised the importance of their own contributions in their children’s school–work transition. While it cost them time and energy, they were not in doubt that their own efforts had a major impact on the employment outcomes for their children:
*But nothing would have happened if it hadn’t been for me… I’ve prepared myself for meetings, I’ve been goal-oriented, I’ve been focused; I’ve been easy to talk with. I’ve been clear about my expectations. And at the same time, I’ve been composed and listening.*

Hence, caregivers seemed to stimulate their children with disorders of ID by having clear expectations about them participating in work life, and they supported them to become independent individuals as much as possible. The participants in our study considered employment as an important part of becoming an adult. Moreover, they believed strongly in the abilities of their children and that they could make a useful contribution to the workplace. In addition, findings from our study also illustrate that caregivers themselves were a driving force during the transition period.

### 3.5. Positive Meetings with Professionals

The participants in our study were very clear about the many obstacles that they had encountered while following their children with disorders of ID during their school–work transition. However, they also mentioned positive meetings with professionals that they believed had eased the transition process and that had contributed to reaching the goal of employment after school years for their child.

Several participants highlighted the importance of meeting “that special person” who is committed to doing a good job. Many spoke fondly of some of the skilled professionals that they met:
*I am really impressed by the people at the supported employment agency, the way they commit themselves to finding strengths in each person. I cannot praise them enough for the work that they do.*

Furthermore, caregivers appreciated it when professionals were accessible, so that they could easily get in touch with them when needed:
*I have a contact person at the supported employment agency. That really means a lot. I’m so very grateful for the contact that I have with her. Really fantastic. I can send her a text message “We need to talk; can you call me?”. And she’s always really great and calls me back immediately. We can talk together.*

Thus, caregivers sometimes experienced meeting with professionals as very helpful in their child’s school–work transition, especially when professionals seemed to work towards the common goal of employment and when they appeared competent and accessible. 

## 4. Discussion

In our study, we explored how caregivers of children with disorders of ID experienced their children’s school–work transition and which challenges they encountered during this process. We also looked into which factors caregivers identified as helpful for the transition. Findings from our study paint an overall picture of the transition process as a time of concern and stress for caregivers, with room for improvement in most areas. This is in line with findings from the study by Strnadová et al. [25], which concluded that parents perceive that more can be done to support young people with disorders of ID during the school–work transition period.

All participants in our study had clear ambitions about employment for their children, and they supported them in their transition by advocating for their rights and by collaborating as best as possible with the support system. Even though we recruited caregivers of young people who had successfully managed the school–work transition, the interviews document that the transition process often appeared random and without an overarching implementation strategy. This seemed to make the transition period extra vulnerable, and a disruption in the daytime occupation for young people with disorders of ID was likely to occur. Despite successful outcomes, only very few of our informants had experienced the transition as smooth. Thus, our findings provide support for previous studies (such as Codd and Hewitt [12]) that describe the school–work transition as a time of stress and uncertainty for parents of children with disorders of ID, rather than for those studies that depict the transition period as a time of personal growth and fulfilment (such as Boehm et al. [17], or Rapanero et al. [18]).

One important point of discussion regarding the findings of our study is how caregivers seemed to reject the disabling discourse regarding their children with disorders of ID. Caregivers expected full participation and inclusion in society for their children, with employment in an ordinary workplace as a natural part of that. This is in line with Parker Harris et al.’s [26] idea that the discourse of welfare is making place for “workfare” discourse in Europe. However, a study by Van Aswegen [27] posits that a medical understanding of disability in which individual deficits and impairments are considered the main causes of joblessness continues to dominate policy documents. The experiences of our participants also illustrate that this discourse is in reality not yet permeating school policy and support systems in Norway. In many instances, a focus on disability still seems to outweigh a focus on capability, and disability is more readily addressed with a disability pension than with a structured pathway into employment. A report by Wendelborg and Tøssebro [28] confirms this finding, suggesting that, in Norway, 88.1% of young adults aged 18–25 with disorders of ID receive a disability pension, while only 3.6% of them are employed in supported employment in an ordinary workplace. This may indicate that the welfare state focuses primarily on securing economic support, rather than on enabling inclusion in society. Nonetheless, the caregivers in our study had higher expectations for their children than merely financial safety, and they emphasised the importance of having a meaningful life. Hence, caregivers seemed to lead the way in altering the existing disability and welfare discourse. Yet, this was not without cost for them, and it demanded effort and courage to make such change happen. Thus, there seems to be a continued need for society to understand and fully embrace the relational nature of disability, where individualised supports may compensate for individual impairments. Providing such supports should be a societal rather than parental responsibility.

In Norway, where this study was conducted, it is not common for parents/caregivers of people with disorders of ID to receive training or education about how they can support their children as best as possible throughout their lifespan. Several of our participants witnessed that they struggled with finding the information that they needed to be able to help their children. This highlights the need for information that is easily available and accessible. A national transition policy with a parental support program and systematic transition planning, as recommended by Scanlon and Doyle [29], could be an effective manner of providing caregivers with the information and support that they need.

Another aspect in this study that deserves further attention is the fact that most caregivers in our study were unimpressed by the school’s focus on post-school outcomes. They often experienced that employment was outside the school’s scope, and in many cases, there was a discrepancy between school’s intentions and reality. Hence, school misses out on the opportunities that they have to prepare students with disorders of ID for future employment. According to a study by Davies and Beamish [21], job training is often absent during school years, and in our study, some caregivers were even left in charge of arranging such work practice themselves. These findings suggest that a lot can still be done in order to enhance the labour market participation for young adults with disorders of ID. Table 2 provides suggestions for measures that could help caregivers during their children’s transition process. The suggestions are derived from our data set, and as such, they stem from caregivers’ personal experiences. Hence, they are not evidence-based, but they reflect areas of need that the participants in our study brought forward.

This study is limited by its sample size regarding the number of participants involved. However, the sample included in-depth interviews with eleven parents/guardians of ten young adults with disorders of ID, with rich descriptions, in which participants highlighted the contemporary problems in this field. Unfortunately, one of the ten recordings failed, and the actual interview was summarised in written text soon after completion. Nevertheless, the other nine interviews were successfully audiotaped and transcribed, and data saturation was achieved. Another limitation of this study is that we failed to collect systematic data on the caregivers’ backgrounds, their educational status, employment status, etc. This lack of thick descriptions of our participants entails a weakness of this study. However, since the purpose of the study is not to generalise, we believe that our findings may still be valuable to researchers and practitioners.

## 5. Conclusions

During the transition period from upper secondary school to working life, caregivers of young adults with disorders of intellectual development (ID) often play an extended, leading role in supporting their children. This article has explored what characterises caregivers’ overall experience of their children’s school–work transition, which challenges they experience, and what contributes to a positive transition. Five key findings were particularly highlighted. With the goal of successful post-school employment, caregivers underlined the importance of ambitions and high expectations on behalf of their children. However, the school’s efforts were often characterized by a lack of preparation for employment during school years. The caregivers had largely experienced a “transition collapse”, and they struggled to navigate the system that was supposed to help their children. Nonetheless, participants identified some success factors that had contributed to a smoother school–work transition for their children, such as the possibility to experience job training during school years and meeting competent professionals who were engaged in the future of their children. Finally, it was emphasised that caregivers themselves were a driving force during the transition period.

## Figures and Tables

**Figure 1 ijerph-20-01892-f001:**
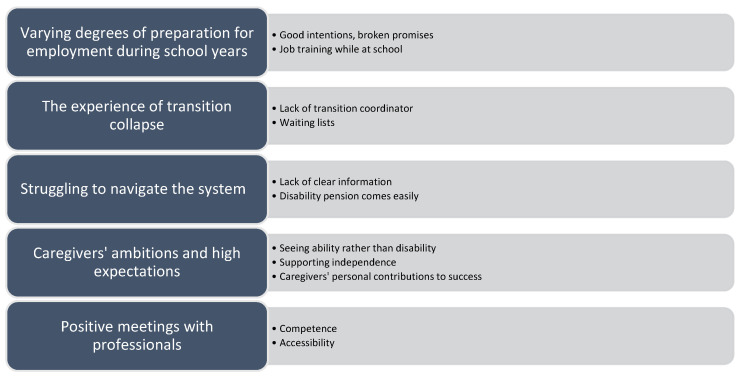
Main themes and subthemes in the data set.

**Table 1 ijerph-20-01892-t001:** Example of data analysis procedure.

Interview Fragment	Code	Subtheme	Main Theme
“It was really difficult to get everyone involved to agree on what kind of plan we were going to have for Simon”	Complex collaborations	Lack of coordinator	Struggling to navigate the system
“In theory, everything was looking good with goals for work training. But then in practice, teachers got sick, they moved, they quit their jobs. And then nothing happened.”	IEP goals for working life skills, but not followed through	Good intentions, broken promises	Lack of preparation for employment during school years

**Table 2 ijerph-20-01892-t002:** Ten suggestions for measures to support effective school–work transitions.

1. Promote the young person’s self-determination, choice making and independence; involve the student actively in the transition planning.
2. Establish a transition team with an individualised transition plan and transition coordinator in order to provide stability and predictability.
3. Hold high expectations and ambitions for the people with disorders of ID; see ability rather than disability.
4. Provide caregivers with clear and easy-to-access information about available supports and the areas of responsibility for each service provider.
5. Start planning the transition early, preferably during lower secondary school.
6. Systems should be person-centred; see the individual’s strengths, needs and interests. What works well for one person may not be right for another.
7. Provide work training during school years, so that students can practice work skills in a real-life context.
8. At the government level, provide sufficient supported employment places to reduce waiting lists. Avoid long gaps between the last day of school and the first day of work.
9. Caregivers need to hear about good things that happen at school and at the workplace; do not only contact caregivers when things go awry.
10. Involve caregivers, so that they know what is happening (even when the person is over 18 years old).

## Data Availability

Due to the possibility of identifying research participants from the interview data, data are not publicly available.

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
