# Peer review of "Caregivers’ Experiences with School–Work Transitions for Their Children with Disorders of Intellectual Development"

_ijerph, 2023, doi:10.3390/ijerph20031892_

Round 1

Reviewer 1 Report

This study is a very interesting one in the field. However, few changes have to be made to reflect the real work that have been done:

- The title should be changed to say "caregivers" instead of "parents" as it turned out to be not all parents participating!

- Questions 2 and 3 could be presented as sub questions to the main question 1 as they both describe what question 1 want to achieve

- I strongly suggest  to change the title to include the "successful transition" or "successfully employed" instead of just "transition" as this what really reflect what is being studied. 

- The participants were not presented fully to the readers, this is really important in qualitative studies because the readers should know more about the people that they are going to hear their stories and experiences. Therefore, I suggest adding a table that present information about the participants ( e.g., age, employment, no. of children in the household,... etc.). The table should present the participants individually like in most qualitative studies. 

At the end, this is a very interesting work and few changes to be made would not prevent it from being published. Thank you  

Reviewer 2 Report

The manuscript provides advance information on transition status that has been missing in the knowledge of intellectual development disorders. The study is significant to care of persons with and their caregivers.

The study gives clear picture of the experiences of the parents regarding school-work transitions for their children with disorders of intellectual development. The work contributes to the important literature needed to improve the well being of people living with intellectual development disorders.

The following comments are made to improve the quality of the study for publication.

Clarity on introduction:

1)      using of ID as abbreviation of disorder of intellectual development or intellectual development disorders against  and intellectual disability (ID)

2)      Consider the social model discourse on definition of disability in the introduction and its impact on employment of persons living with disability

3)      Use of “instead” in a statement which may arise as bias of the authors showing blame. “Instead, insufficiently supportive adult 79 services and what parents describe as “battling against services, lacking trust in services 80 and incompetent professionals” seem to be a cause of negative transition experiences 81 [12] (p.45).”

Clarity on methodology missing the following information:

1)      The study design has information that should be considered as part of sample description.

2)      Recruitment process of participants

3)      Inclusion criteria of study participants

4)      Breaking down of participants including (e.g., number of parents, teachers etc.)

5)      Data collection procedure which can reduce information under ethical consideration

The discussion of the study looks incomplete in the following sense:

1)      missing citation and references of large number of previous studies related to the results of the study.

2)      less support and difference to the results of the study

Compulsory revision of the paper is required to improve its quality before publication.

Reviewer 3 Report

Thanks for the opportunity to review ijerph-2124 Parents’ experiences with school-work transitions for their children with disorders of intellectual development. This manuscript discussed the experience of parents who have young adults with disorders of intellectual development. This manuscript is very well written, although I have several clarifying questions and suggestions for edit. 

Any reason to use the term disorders of intellectual development instead of intellectual disabilities? This manuscript seems to focus on parents of young adults with disorders of intellectual development. You defined that the disorder includes intellectual disability and learning disabilities. Given that your focus is the transition to post-secondary life, I feel compelled if you meant intellectual disabilities. In addition, your references also used intellectual disabilities as a term rather than what you used. If the disorder of intellectual development is a popular term in a specific context, please explain and provide evidence for why. 

Intro

This manuscript reviewed relevant literature that supports the rationale of this study. As written, this manuscript intends to investigate factors associated with parents' experience with children with ID disorders. The introduction and literature review must provide a theoretical or conceptual framework to explore these issues. However, none of the frameworks was able to locate.

The abbreviation for Intellectual development (ID) needs to be used earlier than it is. 

Method 

The case study is an appropriate method to answer the research questions of this study. Specific details of data collection and analysis are enough for future readers to be able to replicate. 

Results 

The authors listed three research questions. 

Q1. What characterizes parents’ overall experience of their children’s school-work transition?

Q2. Which challenges do parents encounter during this transition phase?

Q3. Which factors seem to contribute to a positive transition experience for both the parents and the young adult with disorders of ID? 

How do the themes listed in Figure 1 align with the RQs above? Please be explicit or explain their associations. 

Discussion 

The authors well contextualized the study into the related literature. In 

Table 2 provides meaningful information. To effectively communicate with the readers, please make its format (e.g., bolded) across the suggestions. 

Reviewer 4 Report

This article is suitable to be published with a little additional information, especially about the description of the role that parents have played in this career transition process. Also explain how these parents were given education about making a career transition. Who and how has this been done. The problems they often face should also be explained.
